# Health professionals and scientists' views on genome-wide NIPT in the French public health system: Critical analysis of the ethical issues raised by prenatal genomics

**Adeline Perrot**[1]*, **Ruth Horn**[1,2]

**1** Nuffield Department of Population Health, University of Oxford, Oxford, United Kingdom, **2** Ethik der Medizin, Medizinische Fakultät, Universität Augsburg, Augsburg, Germany

* adeline.perrot@ethox.ox.ac.uk

**Data Availability Statement:** Data for this paper are available from the UK Data Service (DOI: 10.5255/UKDA-SN-855970).

## Abstract

In France, since January 2020, laboratories have started to make available genome-wide Non-Invasive Prenatal Testing (GW-NIPT) beyond the three common trisomies (T21, T13 and T18) at the same cost as standard NIPT. With the possible margins for interpretation of the legal framework and in the absence of clear and updated guidelines, health professionals are left with questions about which type of screening offer may be clinically responsible, morally appropriate, and, at the same time, respectful of women's values and ability to make autonomous choices. The aim of this study is to provide an analysis and understanding of the challenging dimensions of clinical practices in the context of evolving scientific knowledge and techniques in prenatal genomics. In this article, we develop a critical analysis of the arguments and concerns that emerge around the offer of expanded NIPT and are discussed by health professionals and scientists. To achieve this, we conducted qualitative semi-structured interviews with 17 health professionals and scientists from September 2021 to February 2022 and a comprehensive literature review (regulatory, scientific, medical, institutional sources). The results of our empirical research highlight the importance of addressing ethical issues related to the differing quality of counselling, the complexity of achieving informed consent, and the avoidance of harm to pregnant women in the feedback of findings beyond T21, T18 and T13. If there is an increase in the provision of GW-NIPT within the French public health system, it will be essential to promote medical practices that respect reproductive choices of women, support their autonomous decision and their understanding of the limitations and uncertainties associated with GW screening. Further research is required to provide an insight into women's perceptions in order to refine our analysis from the patients' perspective.

## Introduction

Available worldwide [1], non-invasive prenatal testing (NIPT) is a rapidly evolving technology in genetics, increasingly using next generation sequencing (NGS) approaches. In several

**Funding:** RH received a grant from the UK Economic and Social Research Council (ES/T00908X/1). https://www.ukri.org/councils/esrc/ The funder didn't play a role in the study design, data collection and analysis, decision to publish, or preparation of the manuscript.

**Competing interests:** The authors have declared that no competing interests exist.

European countries (e.g. Belgium, the Netherlands, Lithuania, Greece, Cyprus and Italy) [2], laboratories are now providing genome-wide NIPT (GW-NIPT) through the private sector for expanded panels to screen for chromosomal abnormalities beyond the three common aneuploidies (trisomy 21 (T21), trisomy 13 (T13) and trisomy 18 (T18)). In the French public health system, since 2020, expanded NIPT is provided by some health professionals for a wide range of conditions (rare autosomal trisomies, large deletions and duplications $\geq$ 7 Mb) to pregnant women with a probability of a rare condition identified in the fetus, following the combined first-trimester screening (cFTS) (ultrasound examination and biochemical markers). This screening test comes with no additional costs for women and is reimbursed by the public health insurance for the same indications as T21 [3,4], when the maternal serum screening (MSS) test shows a risk score $\geq$1/1000 [5]. Despite some potential clinical benefits of GW-NIPT (screening for an expanded number of chromosomal abnormalities, and the potential to better understand these abnormalities), this screening test also raises a number of ethical issues. In this article, we analyse the views and arguments of French health professionals (gynaecologists-obstetricians, geneticists, midwives, general practitioner) and scientists (medical biologist, researchers in genetics, obstetrics and gynaecology) around the use of GW-NIPT, based on semi-structured interviews (n = 17) and a comprehensive literature review (clinical consent forms, guidelines, regulatory texts, institutional reports, laboratory information leaflets). Our qualitative study, which initially focused on standard NIPT, highlights how the provision of expanded NIPT is the subject of much debate and disagreement within the scientific and medical community. At the same time, the offer is already being made in routine clinical practice by some health professionals (11 of our interviewees) in different regions and clinical settings (midwifery, consultations in gynaecology and obstetrics and genetics).

Our analysis of the interviews show that, besides criticising the clinical validity of expanded NIPT, health professionals and scientists expressed concerns about: the consent for other abnormalities; the complexity of data interpretation for health professionals; the additional needs of training for those prescribing this test on how to appropriately inform women (obstetrician-gynaecologists, midwives and general practitioners); the geographical and clinical settings disparities in the information shared and in the provision of different versions of NIPT ('standard' or 'expanded'); the consequences of an unregulated offer by the private sector for the public health system; the possible impact on women's well-being, and the increase of indications for termination of pregnancy (TOP).

The results of our empirical research highlight the importance of addressing ethical issues related to the differing quality of counselling, the complexity of achieving informed consent, and the avoidance of harm to pregnant women in the feedback of findings beyond T21, T18 and T13. If there is an increase in the provision of expanded NIPT within the French health system, it will be essential to promote medical practices that respect reproductive choices of women, support their autonomous decision and their understanding of the limitations and uncertainties associated with GW screening.

## Context

### Discussions within the international medical and scientific community around GW-NIPT

The growing offer of expanded NIPT gives rise to debates about its clinical usefulness among international scientific societies resulting in differing recommendations. The American College of Medical Genetics (ACMG), for example, does not recommend expanded screening for other autosomal aneuploidies than T21, T18 and T13 [6]. Both the ACMG and American College of Obstetricians and Gynecologists (ACOG) currently point to the lack of evidence of this

screening test with regard to its clinical benefits and accuracy (false positives) [6,7]. The ACMG mentions the difficulties of genetic counselling for conditions with varying degrees of phenotypic expression (e.g. in rare autosomal trisomies). This professional organisation questions the potential consequences (e.g. unnecessary diagnostic procedures, terminations of pregnancy (TOPs)) in the case of pregnancies that are likely to lead to fetal loss (miscarriage). In contrast to these recommendations, other professional societies in Europe, namely in the Netherlands [8] and France [9], are more positive about the possibility of providing findings beyond the common aneuploidies (T21, T18 and/or T13); provided that pregnant women have been informed about the potential risks and benefits of this screening, and given their consent. In its 2020 recommendations, the Association of French-speaking cytogeneticists' working group on NIPT [9] recognises that there is no consensus and sufficient data in the literature on what to do when abnormalities other than T21 are discovered through GW-NIPT.

More broadly, in recent years, there has been a lively debate within the scientific and medical community about the clinical utility of reporting NIPT findings based on whole genome sequencing (WGS). While some studies [10,11] question the clinical interest and performance of expanded NIPT (e.g., false positives generated by the detection of mosaic confined to the placenta), others highlight [5,12] its value to provide information (e.g. for rare autosomal trisomies and for large duplications or deletions) as well as diagnostic clarification (e.g. for some chromosomal rearrangements or false positive results for T21, T18 and T13).

The French debate around expanded NIPT reflects these polarised views in the international scientific and medical community. Before developing a critical analysis of the concerns that emerge and are discussed by health professionals and scientists, we will present the French context where expanded NIPT is offered in certain clinical practices, and its usefulness is discussed.

## Discussions on the report of findings and the legality of GW-NIPT in France

Following the implementation of standard NIPT, in January 2019, and its coverage by the health insurance for T21 (often including testing for T18 and T13) [3], the French laboratory Cerba started to provide expanded screening beyond the common aneuploidies in January 2020 [13]. This test is based on Illumina's 'VeriSeq NIPT Solution V2' [14] which extends the screening possibilities for a wider range of chromosomal abnormalities. These include trisomies 2, 8, 9, 14, 15, 16 and 22, large duplications and deletions. Currently, sex chromosome aneuploidies and microdeletion syndromes ≤ 7 Mb are not screened in France. The large size of some chromosomal abnormalities (duplications and deletions ≥ 7 Mb) is used as an argument for the feasibility of detection and the possibility of classifying a chromosomal variation as 'potentially pathogenic' [15].

A similar test is now also marketed by a second laboratory, Eurofins Biomnis [16]. As for standard NIPT, this expanded test version is available either privately or can be reimbursed as a second-tier test for the same indications as T21 (MSS risk score ≥1/1000). A requirement for prescribing expanded NIPT is the informed consent of women. In case of a positive result following GW-NIPT, further invasive tests (amniocentesis, chorionic villus sampling) followed by a confirmatory chromosomal microarray should be offered. At present, there is no estimate of the number of health professionals and the frequency with which they propose this new version of NIPT in France. Before discussing the main concerns that emerge around this provision of GW screening in clinical practice, we can already note two major points of debate in the French public health system.

First, there is no consensus among health professionals on whether findings from expanded NIPT should be considered 'primary' or 'additional' findings. In the first case, data sought are

related to the initial indication of a higher probability of having a child with a genetic condition following cFTS. According to this view, the use of GW-NIPT leads to actively generating information about other chromosomal abnormalities, and the extended findings are part of the GW-approach. In the latter case, 'additional' findings are unrelated to the initial indication for the screening, but may reveal information about a genetic condition. They can be either unexpectedly found or intentionally sought, and should be returned to women if they have previously given their consent to receive such information. The Law of 2021 on Bioethics (art. 16, 16–10, II, 4˚) [17], dealing only with discoveries made incidentally, does not appear to address directly the question of intentionally sought information ('secondary findings') and leaves uncertainty about whether it should be communicated to women (or not). The working group on additional findings at the Biomedicine Agency is expected to clarify this point in the near future.

Second, another point of concerns is whether it is within the scope of the law to offer NIPT for chromosomal abnormality other than T21. While the Haute Autorité de Santé (HAS) recommendations, as well as the decree of 14 December 2018 regulating prenatal testing, regulate T21 without mentioning T18 and T13, none of the regulatory texts explicitly limit NIPT to the detection of T21 [18]. The broad category of 'condition' used in the decree of 14 January 2014 on antenatal diagnosis (Art. R. 2131–1.-I.) [19] does not specify the scope of aneuploidies to be screened for and leaves room for interpretation of the 'condition' that can be looked for in routine prenatal care. Legal scholars have been arguing that the Article L2131-1.-V. reformed by the Law of 2021 on Bioethics could also be understood as an opening for expanded screening, referring to 'the possibility of detecting a particularly serious condition in the embryo or foetus (. . .) or a condition that may have an impact on the future of the fetus or unborn child' [17]. Nevertheless, this recent law does not clarify the applicability of the current legal framework for screening practices, referring only to tests for 'diagnostic purposes' [18].

With these possible margins of interpretation of the legal framework and the absence of clear and updated guidelines, health professionals are left with questions about which type of screening offer may be clinically responsible, morally appropriate, and, at the same time, respectful of women's values and ability to make autonomous choices. This is why we conducted this study, to provide an analysis and understanding of the challenging dimensions of clinical practices in the context of evolving scientific knowledge and techniques in prenatal genomics. Health professionals and scientists may have plural perceptions and arguments about how to 'act ethically' in their daily practice. Just because GW-NIPT is available and not regulated does not automatically make the offer an acceptable option. Health professionals and scientists are hesitant not only about the clinical relevance of the screening options made available to them by laboratories, but also about their ethical implications: is GW-NIPT good practice? And, does it benefit pregnant women or not?

## Method

This paper is part of a wider comparative study combining literature review and empirical research (semi-structured interviews with different stakeholders, including pregnant women/ couples and health professionals) to explore the ethical issues arising from prenatal genetics and genomics in England, France and Germany.

First, we conducted a comprehensive literature review focusing on public discourses and regulations about non-invasive prenatal testing (NIPT) in the three countries [20]). Between December 2020 and April 2021, we reviewed approximately 250 sources in legal and regulatory texts; public reports of national ethics committees and professional bodies; parliamentary debates; academic literature in prenatal genetics and genomics, Bioethics, Social Sciences,

medical and daily press. We searched the databases of Cairn journals (Humanities and Social Sciences), Google Scholar, PubMed and SAGE journals. We focused on literature, since 2011, when standard NIPT became first available in the private sector before several countries, including France, decided to fund it within their public health system. The sources reviewed in this article focus specifically on discussions related to the development of GW screening techniques and the feedback or disclosure of additional data. They cover international scientific articles, international guidelines, French regulations, hospital and laboratory documentation as well as French parliamentary report.

Second, in order to better understand attitudes and practices of French health professionals and scientists regarding NIPT, we conducted 17 semi-structured interviews in French from September 2021 to February 2022. Among the health professionals, we interviewed 4 obstetrician-gynaecologists, 6 clinical geneticists, 5 midwives, 1 general practitioner and 1 medical biologist. 11 of them are also scientists. The inclusion criteria for the study were healthcare professionals and scientists practising in France who had experience of offering NIPT, including providing genetic counselling to pregnant women, and/or of conducting research around NIPT. The exclusion criteria were health professionals and scientists who are not involved in the offer or development of NIPT. The interviews lasted between 30 and 60 minutes and were conducted online via MS Teams. We used a thematic guide in order to gain a better insight into professionals experience with NIPT including their views about benefits and difficulties regarding the test, the discussions taking place when giving information or returning results to women, and their views about future developments of the test. Although the interviews did not explicitly focus on expanded NIPT, the topic emerged as a cross-cutting theme. At the time of our interviews, standard NIPT had already been introduced in the public health system, for about 3 years, and many health professionals (11 of our interviewees), including midwives in hospitals and private practices, were already offering expanded NIPT as part of routine clinical care. During data collection, we noted how the subject of extending the screening offer is currently at the centre of debates in France, and how health professionals and scientists express concerns about what they perceive as adding complexity to the counselling practices for women/couples in antenatal care.

Ethical approvals have been obtained from Oxford Central University Research Ethics Committee (R64800/RE001) and the French Inserm Ethics Evaluation Committee (Inserm Ethics Evaluation Committee (CEEI)/Institutional Review Board (IRB): Avis n˚21–82).

Prior to the interviews, participants received a participant information sheet. On the day of the interview, consent was obtained to conduct, record, and transcribe the interviews; to use anonymised quotes in scientific publications; to store de-identified transcripts, and to deposit these in the UK Data Archive. Consent was obtained online by reading the consent form out aloud and asking the interviewee whether they agree or not. A copy of the consent form signed by the interviewer was then emailed to the interviewee for their records.

The interviews were coded and cross-coded by the three researchers involved in the larger research study, and analysed using NVivo software. We wrote memos to develop the analysis and to see how themes emerged through constant comparison with the data. Thematic coding revealed sub-themes related to the cross-cutting debate on extension of screening using GW approaches: technical limitations and the clinical utility of screening for other chromosomal abnormalities; management and feedback/disclosure of findings; format and achieving of informed consent; quality of counselling; training of professionals; relationships between the public and private sectors; scope of expanded NIPT; extension of indications for termination of pregnancy (TOP) and impacts on women's well-being.

We are investigating how health professionals and scientists are formulating normative views about what should be done in the context of laboratories making available expanded NIPT at the

same cost as standard NIPT, i.e: whether or not GW-NIPT should be offered to pregnant women? What should be the scope of this screening if extended? Is there a need for regulation?

## Results

While the interviews with health professionals and scientists highlight some potential benefits of GW-NIPT (screening for an expanded number of chromosomal abnormalities, and the potential to better understand these abnormalities [21]), they also identify a range of concerns.

### The importance of additional professional training and availability of appropriate resources

In the French context, one of the major problems raised by the interviewees is the need for additional training of health professionals prescribing GW-NIPT. Some of the interviewees note disparities in the information shared with women/couples according to the services as well as according to the professionals (gynaecologists-obstetricians, midwifes and general practitioners) in charge of prescribing and delivering information about NIPT. As described by this geneticist, the difficulties are likely to be exacerbated if the screening moves from a standard NIPT to an expanded one:

'And we realise that just on T21, (. . .) the information is difficult to give, that it is less well given by gynaecologists than it can be given by geneticists or genetic counsellors. And so we can imagine that gynaecologists who give this fragmented information on finer chromosomal abnormalities, potentially detectable, will do a less good job. (. . .).'(Geneticist 1)

This raises the question of resources for training of health professionals who prescribe the test to keep up to date with information about new genetic and genomic technologies [22,23]. With regard to geneticists and genetics counsellors, the issues are related to staff availability and the need to train them as screening techniques evolve at a fast pace:

'We took the time to train everybody [in his/her team] and we have two genetic counsellors, but (. . .) there are not enough counsellors.' (Obstetrician-gynaecologist 1)

In this section, we have seen how extending the offer of NIPT beyond the three common trisomies can be particularly challenging in a context of limited counselling resources, although new courses for genetic counsellors are currently being opened as part of the France Genomic Medicine Plan 2025 [24]. This highlights the challenge to increase the number of genetic counsellors in France and to provide training for HCPs in order to support women's decision-making.

### Challenges of obtaining women's informed and autonomous consent

Interviews conducted with professionals underline the complexity of achieving informed consent when screening for chromosomal abnormalities other than T21, T18 or T13. They explain this complexity notably through the different ways each of the laboratories carries out expanded testing in France and how they seek consent. While both main laboratories provide the explicit option to opt-in for other information than the common aneuploidies, Cerba provides a precise list of additional chromosomal abnormalities (trisomies 8, 9, 12, 14, 15, 16 and 22; large deletions and duplications $\geq$ 7 Mb) and Biomnis proposes to return results "for any rare abnormalities that may be identified provided the patient consented to this"; it is however not specified which abnormalities are screened for [25,26]. Health professionals and scientists in our interviews express concerns about the lack of harmonisation in antenatal care offered

through the public sector. They point out the different information women may receive depending on the laboratory that carries out the analysis:

'(. . .) it is absolutely necessary to persevere and (. . .) to impose that the consent be the same for everyone, and not [to have different] consents by laboratory (. . .), otherwise, it's an open door to everything and anything.' (Obstetrician-gynaecologist 2)

The need for standardisation of tests and consent forms have led some clinicians to modify and adapt the consent forms used in their own practice. One of them added an explicit option (yes/no) for the search for other abnormalities than T21, T18 and T13, whereas initially the laboratory form only provided for an opt-in choice:

'So, I reformatted it [the consent model] (. . .) in my hospital and (. . .) there's a specific box with the question asked as directly as possible (. . .) and a yes/no answer to be ticked off, saying that, well, if it's not ticked off, it means it's no.' (Geneticist 2)

This geneticist wants to give pregnant women the option of explicitly consenting or refusing an expanded testing, rather than only giving the possibility of agreeing or complying to the screening proposal. However, giving women the option to consent or not to the search for other chromosomal abnormalities can come as a surprise or an unexpected choice, as described by this midwife:

'(. . .) the [laboratories] are suggesting that you go and look at something else potentially. This can make patients feel uncomfortable, I think, because they don't expect to be given a choice.' (Midwife 1)

This midwife expresses her concern that expanded NIPT could increase the burden of choice for women and complicate the consent process. Decisions about the format of consent and the information to be shared during pre-test counselling may need to be harmonised to facilitate voluntary and uniform consents from women.

In some clinical situations, the issue of consent may have become problematic in that the feedback or disclosure of additional findings was not subject to the prior consent of pregnant women (or only *a posteriori*). This problem of not respecting the right of women to consent was highlighted and discussed at the 2021 Fetal Medicine Days in Marseille:

'There is no consensus [GW-NIPT]. We [health professionals] discussed this in Marseille. (. . .) There was a big debate which was quite heated, i.e. there were some who criticised certain laboratories for having, despite what the women had asked for, produced reports saying: "we're not giving you the result [without your consent] but it would be good to return [the consent form] by ticking that you want the incidental findings", which is a way of forcing consent.' (Obstetrician-gynaecologist 2)

The problem raised in this interview is that the data reported should be related to the information given before undergoing the test and to what the woman has consented to. Women should not be encouraged to agree to seeking additional information. This previous approach may have carried a risk of harming pregnant women by inducing the necessity for consent. The problem appears to be being addressed with the preparation and dissemination of new consent forms, in particular by one of the Greater Paris University Hospitals (APHP), which now includes the possibility of seeking abnormalities other than T21, T18 and T13. It is crucial

to improve pre-test information and consent on the basis of what has been discussed during the counselling, as one professional point out:

'If the patient has signed up for 13, 18 and 21, we don't give her anything else. (. . .) And, this is very difficult for the cytogeneticists because they see the abnormality. (. . .) But the way to avoid all that is to do as the Belgians do: it's to switch to GW testing for everyone in the first instance and at least explain to the patients why this test is being done, whether they want to do it or not.' (Obstetrician-gynaecologist 1)

This suggestion to provide GW-NIPT as a first-tier test is intended to clarify the consent process. However, this active search for genomic findings is at the centre of the discussions. Are they clinically relevant, i.e. actionable for the management of pregnancy [27]? As mentioned above, there is currently no agreement on how to define actionability in view of the multiple possible interpretations from the perspectives of physicians, patients or families. Even more though in the prenatal context, the category of 'actionability' needs to be rethought because the measures that can be taken are rarely therapeutic (*in utero* intervention) but preventive (TOP).

## Lack of regulation around expanded NIPT

To date, there is no consensus among health professionals about the provision of expanded NIPT. Some believe that it exceeds regulatory limits and others think that it is not unlawful even though it is not included in the regulatory framework, as illustrated by these two interviews:

'If the Agency of Biomedicine has been letting us prescribe these tests for two years now, it is because it is legal.' (Geneticist 3)

'(. . .) it's not because it's not written in a law that it's illegal, and that's the other problem. (. . .) It is part of the prescription of genetic anomalies that we have the right to do for any patient, simply the framework of this practice is not legislated.' (Obstetrician-gynaecologist 1)

Beyond the different interpretations, professionals underline the importance of regulating the expanded offer of screening from private laboratories:

'At some point, we have to put in place a legal framework to protect people (. . .) there's the whole question of obstetrician colleagues who are not trained [in genetics], who find themselves doing things they're not comfortable with (. . .). In fact, at some point, physicians cannot be taken hostage by the private sector to make a diagnosis that they do not know (. . .)' (Obstetrician-gynaecologist 2)

This obstetrician-gynaecologist draws attention to the extent of the data available and the risk that health professionals, who are not adequately trained, may have difficulties in interpreting it. The increased offer initiated by the laboratories raises concerns among health professionals and scientists who call for the organisation of multi-disciplinary meetings to determine a common position within the public sector:

'There are quite a few discussions about the extension of NIPT (. . .) because the private sector has started to extend NIPT beyond what was done in the public sector (. . .). It is absolutely necessary that we arrive [at a position] in France that is very consensual (. . .) because,

for the moment, we are, in fact, in a rather delicate situation at the moment, somewhat generated by the private sector, which has rushed things a little with regard to what could be proposed afterwards.'(Geneticist 4).

In the face of accelerating prenatal technological developments, some professionals refer to the Belgian Society of Human Genetics [28] that has agreed on a list of incidental findings classified as 'valid' (technically and with validated evidence on the associated phenotype) and that can be reported at a national level. However, in France, it seems to be more difficult for professionals to reach an agreement on GW-NIPT findings. Also, it is interesting to note that in the French context, the intention of physicians to offer the expanded screening as a publicly funded test has resulted in efforts to develop manual sequencing technologies within hospitals. Nevertheless, private laboratories have an automated method and CE-IVD validated tests, and therefore perform most of the tests. With this centralisation of tests by two laboratories (Cerba and Biomnis), some health professionals and scientists are concerned about the uncontrolled development of GW-NIPT, repeatedly referring to the risk of 'escalation'. There is a perception here of a technology whose progression is not regulated and whose prescription seems both easy ('simple blood test') and complex to manage:

'So, we start to go into debates, it becomes complicated and difficult for clinicians to know where to stop. (. . .) Does she have [the patient] to see a geneticist before she does a NIPT? Will the geneticist say: "Wait, (. . .) if I had been consulted, I would have given information and she might not have ticked that box?" We are in an escalation that for the moment seems out of control.' (Obstetrician-gynaecologist 3)

This interpretation of GW-NIPT offer is about the potential influence of the physician over the way women make choices about their pregnancy. There is indeed the fear of inducing a routine offer of expanded NIPT, which women find difficult to refuse:

'(. . .), there's a bit of an escalation. I think, it's a bit difficult because the patients [will be thinking]: 'my physician is suggesting it [expanded NIPT] to me, so it must be interesting'. And so, if it's suggested to me, there must be a reason and so I'm going to follow the medical advice.'(Obstetrician-gynaecologist 2)

This interview refers to the risk that women could feel pressured to accept GW-NIPT because it is offered by the health professional. The question of whether women's reproductive autonomy could be compromised when offering NIPT as part of routine clinical care ('routinisation') is not new [29,30]. However, several studies have shown that women make informed choices based on their own values regardless of whether NIPT is offered in routine clinical care or not [31–33]. These results will need to be re-evaluated in the light of expanded NIPT and hence expanded screening options.

## Potential impacts on women's well-being

Some health professionals and scientists raise concerns that the potential harm of expanded NIPT could outweigh its benefits. They highlight the potential psychological and psychosocial harms to women and their families when reporting information related to other chromosomal abnormalities for which the test does not perform with high accuracy [23]. This is the case of this obstetrician-gynaecologist who wonders about the clinical relevance of the search for chromosomes rearrangements that are likely to cause miscarriages or could be false positives. She/he decided not to offer this extended version during the screening examination:

'(...) our objective as physicians, above all, is not to harm our patients and that, at some point, [we risk] inducing such significant stress in our patients (...). We also know that [in] all prenatal diagnosis (...), there are studies on ultrasounds with minor variations and there are psychosocial consequences, even on the long-term development of the child and the family.'(Obstetrician-gynaecologist 2)

In this interview, the professional develops her/his argument around the potential familial implications and the development of the fetus, which may be impacted by the provision of uncertain medical information.

Among the other concerns about harms to women's well-being, health professionals are also worried about the feedback or disclosure of additional findings on women's health (such as carrier status or malignancy in the mother), even if these data are not currently reported in France, as is the case in Belgium:

'So, I think, we're in a context where progress is generating [a provision of] care that didn't need to exist in fact. (...). In other words, (...), we can say to ourselves: "in any case, [the offer of the test] will reassure them". But, the anxiety generated, in my opinion, can sometimes be catastrophic.' (Geneticist 4).

This cautious approach aims to protect women from information that could potentially be harmful. If this is related to the principle of non-maleficence in health care, it also raises questions about what women do or do not want to know, despite the implications this information may have on the experience of their pregnancy.

Some interviews suggest that the pre-selection by scientists and health professionals of the relevant chromosomal abnormalities to be screened may partly address concerns about the anxiety that could be generated. Indeed, some professionals believe that the defined limit in the search for aneuploidies, large deletions and duplications, may precisely prevent the risk of causing unnecessary stress in pregnant women avoiding reporting more benign or uncertain data:

'It's really on the experience acquired in classic cytogenetic analysis in the past 30 or 40 years, (...) that we selected these 7 chromosomal pairs for which we say: "ok, we're going to report the suspicions of aneuploidy because there is a possibility of medical action". On the contrary, for the others [trisomies], we considered that the risk was very, very low. (...) So we considered that it was medically irrelevant to give basic stressful information and generate invasive procedures most often for nothing.'(Medical biologist 1)

In this approach based on a selection of abnormalities to be screened for, health professionals and scientists examine the available scientific data to determine which chromosomal abnormalities may be of clinical interest and limit the anxiety generated in women. If it seems to be supportive of pregnant women's autonomy choosing whether to undergo expanded NIPT or not, this approach implies a certain degree of paternalism [34] in order to delimit the panel of abnormalities that is subject to GW screening.

Finally, we can note here that it will be essential to develop counselling methods that address the potential impacts on women's well-being in specific clinical settings, while promoting meaningful choices [35] that may mitigate anxiety. Indeed, studies have already stressed how the possibility to make an informed choice can limit women's anxiety before the test is carried out [36].

## What should be the scope of GW-NIPT?

Discussions among health professionals and scientists also focus on the limits to be placed on screening for rare chromosomal abnormalities, particularly with regard to their clinical value. The use of GW-NIPT raises questions of how far to go or where to stop:

> [NIPT raises questions about] how far we can go regarding the pathologies that we can detect (. . .). So, we know that the easier a test becomes, the more likely it is to (. . .) be prescribed by people [health professionals] who are not necessarily very careful, not very attentive, who do not necessarily know the pathologies very well and so on. (Geneticist 5)

Concerns include the limits to be established in the offer of the test and the challenge of training staff in relation to the complexity of interpretation of rarer chromosomal abnormalities than T21, T18 and T13 [37]. It raises the question of how to decide what information is relevant or not to feedback to pregnant women/couples? These concerns are linked to the technical limitations of GW-NIPT (i.e. false positives). In these circumstances, genetic counselling is made difficult as is apparent from our interview with a geneticist:

> 'So when we did just prenatal diagnosis of T21, it was quite simple, well, T21, T13, T18. And now, there you go, we see some complications.' (. . .) There, so with the extension (. . .) we see that the results are not easy to manage, that we're going to have perhaps false-positives, etc.' (Geneticist 4).

Health professionals prescribing expanded NIPT, such as midwives, are likely to refer patients with positive results to geneticists to ensure that patients receive the appropriate information. The scope of NIPT is also questioned in this interview with another geneticist. He/she underlines the issues this will raise in relation to screening for other genetic variants, and, in particular, for intellectual disabilities:

> 'The question will be: "Where do we stop? Why should we screen more?" If we start doing GW screening or finding deletions of the long arm of chromosome 7, why would we do that more than screening for mutations that causes intellectual disabilities?' (Geneticist 2)

We can see here that the boundaries to be set on the extent of the NIPT in clinical care is still the subject of an important reflection in the medical community in France. This is also the case in Canada where health professionals seem to agree on the feedback of findings for fetal aneuploidies and monogenic diseases, but not for non-medical conditions and risk predisposition information [38]. Health professionals will have to find an agreement at a national level because of the disparities between different cities, regions or clinical settings, some of which are more inclined to broaden the indications further than others:

> 'You can have 10 [different multidisciplinary centres for prenatal diagnosis] in France and I think you'll have 10 answers which are not completely identical. We know very well that we in [medium-sized city] legislate more or less in a direction. You go to Paris, to Necker [hospital] and they will say "yes" to everything. (. . .) They have a policy [that is adapted to] the population of the 14th and 9th arrondissement that is much more demanding, and they will tell you that where [there is a risk of the mother not bonding with her child], it is better to terminate the pregnancy.'(Midwife 2)

These variations in geographical and clinical contexts add complexity to the provision of antenatal care, which is supposed to be uniform within the French public health system. In this respect, the possible extension of the offer raises the question of what kind of screening should be offered to all pregnant women and not only to those who have better access to information and to trained professionals willing to offer NIPT expanded.

## Expansion of indications for TOP?

The concern about the scope of screening is also related to the issue of the possible expansion of indications for termination of pregnancy (TOP). In France, since March 2022, 'voluntary' TOP has been extended to the end of the 14th week of pregnancy instead of the 12th week [39]. At the end of this period women have access to a medical termination of pregnancy at any time (resulting from the decision of a multidisciplinary team). This raises the question of whether expanded NIPT is likely to lead to an increase in the number of TOPs, especially with the extension of the period of 'voluntary' termination?

> 'Afterwards, there is also the whole question of, which is perhaps more about the type of chromosomal abnormalities that would be detected because, in fact, there are chromosomal diseases that will not always be serious, [and] not always admissible for a TOP.'(Geneticist 4)

While decision-making dilemmas around TOPs for medical indications already arose in the context of standard NIPT, the expanded version is likely to broaden the range of questions that will be posed to professionals:

> 'Is it possible that, after an abnormal result, we will grant requests for medical TOP for all types of pathologies? So, this raises the question of how far we can go in terms of indications, and there are differing points of view because, well, there are the experiences of families to take into account. But, we also know that sometimes there are people who are extremely vigilant about, well, the state of health of their future child. (. . .) So that's quite difficult and so how far we can go in [meeting] the requests of the couple, of the pregnant women?'(Geneticist 5)

Through this interview, we see how women's perception of the seriousness of genetic abnormalities is re-interrogated in light of what health professionals themselves consider a medical indication for TOP.

These contrasting views between patients and health professionals about TOPs also appear in a study conducted by the French Council of State in 2018 [40], in this case, on the subject of 'voluntary' TOP [39]. The Council of State questions the ethical issues raised by GW-NIPT in the context of genomic sequencing: 'There is a risk that couples choose abortion (. . .) based on a subjective and irrational understanding of the seriousness of a genetic abnormality which they are told to be present, (. . .)' [40]. Questioning the rationality or thoughts of women/couples in making choices about their personal circumstances and projects is not new. It might be reinforced in the context of expanded information for rare chromosomal changes whose severity is difficult to establish because of the variability of expression (as is also the case for the three common trisomies). This external political view tends to question the reasoning of women/couples who may develop a different perception to that of health professionals on how they envisage their family plans and the support they have access in order to meet the needs of a disabled child [41]. Whatever decisions are made about whether or not to extend NIPT, it seems that women's perspectives around their reproductive choices will need to be taken into

account in order to develop a more patient-centred approach to antenatal care in the French socio-cultural context.

## Discussion

Through the arguments mobilised during the interviews, French health professionals and scientists express different positions regarding the provision of GW-NIPT in France. These positions reflect the current context of increasing availability of GW-NIPT in routine clinical care without a clear regulatory framework or regularly revised recommendations from health authorities. This results in differing clinical practices regarding the choice given to women to carry out expanded NIPT or not.

The findings of our study show the need to clarify the provision of NIPT ('standard' or 'expanded') to ensure that the offer is equitable between women in different regions and clinical settings. To mitigate these disparities in the screening offer, equal access to information and training resources for professionals may be of benefit. This may help reduce knowledge gaps and harmonise practices. If GW-NIPT is to become more widespread in the future, it seems important that women receive the same objective information across the country, regardless of the health professionals' personal values and views. Women should not feel pressured but be enabled to make a decision based on their own beliefs. Therefore, it seems crucial to consider the development of standardised information in the form of booklets or online educational tools for patients [42]. This could facilitate access to detailed written information, in addition to verbal information provided during the consultation.

As our empirical data has shown, particular attention should be paid to pre- and post-test counselling to discuss potential results and technical limitations of NIPT (test failures, false positive, etc.) in GW analysis [43]. This should help to manage the expectations of women and avoid deceptions or misinterpretations about the scope of the test, and ensure that the difference between this screening tool and a diagnostic test is understood [44].

In addition, expanded GW-NIPT provided within the public health system raises the question of the level of information [45] that women should expect to receive for consent to be valid. For example, in the Biomnis form (laboratory), the consent is obtained without mentioning other potential abnormalities that may be identified [46]. While it is important to provide information, an overload of information could undermine the ability of pregnant women to make autonomous choices [47] and should be avoided. This will involve reducing the complexity of the information provided during the counselling process while providing the essential elements to support decision-making.

Finally, it will also be important to weigh the risks and benefits of offering expanded NIPT to pregnant women and how to avoid harm, in particular the stress and anxiety that may be generated by reporting findings for abnormalities for which the test accuracy is debated. The impact on women's well-being might depend on the circumstances (e.g. waiting for test result, receiving a low/high probability result) and the interaction between the physician and the patient (e.g. possibility for patients to get answers to their questions, compassionate attitudes of professionals). With the current developments in prenatal genomics, it will be important to look at whether expanded NIPT modifies women's perception of risks: does it increase the fear that a condition can be detected in the fetus or does it, on the contrary, reinforce women's earlier reassurance and enhance the possibility to project themselves into the pregnancy and bond with the fetus/future baby? A study (Lewis, *et al.* 2016) has identified women's tolerance of elevated anxiety in accepting NIPT in order to obtain more information and to be reassured about their pregnancy [48]. Nevertheless, they discovered cases of prolonged or additional anxiety in women at intermediate risk (cFTS) following a negative NIPT result [48]. It shows

that the health risks for the fetus and the woman require special attention and may involve providing further counselling and support in specific situations. These results (Lewis, *et al.* 2016) should be interpreted with caution and cannot be transposed to the context of an extended screening test given the multiple conditions tested for.

## Conclusion

The increasing provision of GW-NIPT in the French health system is redefining the different responsibilities of health professionals in the context of a complex test offer. It raises questions about the appropriate counselling that this requires to ensure informed consent and avoid harming pregnant women. Expanded NIPT raises the broader issue of how to preserve an environment of trust and reliability in the clinical setting in the absence of specific regulations and guidance. Guidelines should be developed to deliver high quality antenatal care that promotes reproductive choice and respect women's values, beliefs and preferences. Pregnant women should be able to express their views about the development of GW screening and make their own decisions based on their own rationality and experience.

## Limitations of the paper

In this paper, we have highlighted different perceptions of French professionals on what they consider to be best practice in the context of expanded NIPT. Women themselves may have different opinions about what is 'good' screening practice. This paper does not allow to make any claims about what women think of expanded NIPT. However, some studies show that the majority of women are choosing access to results beyond the three common trisomies after pre-test counselling: in the Dutch TRIDENT-2 study, 74.2% of women chose to have additional findings reported [21] and pregnant women are willing to accept a less accurate test to obtain more information on fetal chromosomal status [49]. This raises the question of whether physicians should decide what tests to provide or whether, and if, to what degree, this decision should be left to women/couples?

Further research is required to provide an insight into women's perceptions in order to refine our analysis of the ethical issues from the patients' perspective.

## Author Contributions

**Investigation:** Adeline Perrot.

**Supervision:** Ruth Horn.

**Validation:** Adeline Perrot, Ruth Horn.

**Writing – original draft:** Adeline Perrot.

**Writing – review & editing:** Adeline Perrot, Ruth Horn.

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
