## [Decision Letter · Decision Letter 0]

5 Sep 2022

PONE-D-22-21988Health professionals and scientists’ views on genome-wide NIPT in the French public health system:

Critical analysis of the ethical issues raised by prenatal genomicsPLOS ONE

Dear Dr. Perrot,

Thank you for submitting your manuscript to PLOS ONE. After careful consideration, we feel that it has merit but does not fully meet PLOS ONE’s publication criteria as it currently stands. Therefore, we invite you to submit a revised version of the manuscript that addresses the points raised during the review process.

We look forward to receiving your revised manuscript.

Kind regards,

Antonio Simone Laganà, M.D., Ph.D.

Academic Editor

PLOS ONE

Journal Requirements:

Additional Editor Comments:

The topic of the manuscript is interesting. Nevertheless, the reviewers raised several concerns: considering this point, I invite authors to perform the required major revisions.

Reviewers' comments:

Reviewer's Responses to Questions

**Comments to the Author**

1. Is the manuscript technically sound, and do the data support the conclusions?

Reviewer #1: Yes

Reviewer #2: Yes

Reviewer #3: Partly

2. Has the statistical analysis been performed appropriately and rigorously? 

Reviewer #1: Yes

Reviewer #2: N/A

Reviewer #3: N/A

3. Have the authors made all data underlying the findings in their manuscript fully available?

Reviewer #1: Yes

Reviewer #2: No

Reviewer #3: No

4. Is the manuscript presented in an intelligible fashion and written in standard English?

Reviewer #1: Yes

Reviewer #2: No

Reviewer #3: Yes

5. Review Comments to the Author

Reviewer #1: Dear authors,

congratulations for your paper. NIPT is the future of prenatal diagnosis, but Is crucial to understand that information about is pivotal

I think you paper is interesting, I would suggest to try to ameliorate the presentation of the result in a more schematic way, moreover to add a table with a summary of findings

I would recommend to go even more thought the crucial role of women awareness of the meaning of this test, is has been demonstrated worldwide that women understanding of this test is still quite confusing, the words diagnostic and screening are still often confused this pose enormous risk for health care professional, for this point I suggest to read and cite:PMID: 33111167

best regards

Reviewer #2: This is an interesting article reporting health professionals’ views on genome-wide NIPT in the French public system. While the results are interesting, major flaws should be addressed especially when it comes to the methodology, quotes translation, and English academic writing.

For the methodology, the authors state that they analyse the views of French HPs around GW-NIPT, based on semi-structured interviews and a comprehensive literature review that includes clinical consent forms, guidelines, regulatory texts, institutional reports, laboratory information leaflets. While they report their findings from the qualitative data, the comprehensive review part is missing. So, it is not clear if the authors conducted two separated studies (empirical and conceptual analysis based on comprehensive review) and that they are trying to combine in one study and in such case, they should describe how the comprehensive review was performed, the inclusion and exclusion criteria, what kind of studies were included, etc. or if they are referring to the literature from diverse sources to better interpret their findings.

The translation of the quotes needs to be revised as some sections seem to reflect a literal translation and the written English should conform with the academic English writing throughout the manuscript. Many sentences are structurally unclear, rendering therefore the meaning and the understanding very complex and unachievable. Here are few examples:

Line 360: “…by the private sector to make diagnostic returns that they do not know. What does diagnostic returns mean? The results?

Lines 370-371: “…It is absolutely necessary that we arrive in France at a situation that is, in fact, very consensual”. The translation is unclear.

Lines 551-552: “Questioning the rationality of women-couples in making choices about their personal circumstances and projects is not new”. What does rationality mean in this context?

Line 437: we can say to ourselves. This is a sentence that should be reviewed.

Lines 446-447: “anxiety that could be generated in women by GW-NIPT”. Please review.

Line 483: “How to decide what information is relevant or not to feedback to pregnant women”. Please review.

Lines 525-526: “Is the test likely to lead to an increase in the number of TOPs whether?” Please review.

Line 530: “not always admissible for a TOP”.

Line 534: “questions that will be posed to professionals”…please review. And the list goes on…

Introduction

p.4 line 73: The authors state “despite clinical benefits of GW-NIPT” without mentioning what are these. Please nuance.

Line 74: this screening test is largely driven by industry. So what does this mean and how it does impact its use and implementation?

Context

Why not give context about TOP in France? Is it legal? Prohibited at some point in time during the pregnancy? The authors only briefly discuss TOP as a theme that was developed in their interviews. It would be of benefit to the reader to have a glance about the situation of pregnancy termination in France.

Methodology

Lines 201-204: Many details are missing from the methodology used to collect the data. For instance, how the themes were developed and how were they validated? In what language were the interviews conducted? How the translation into English was done and validated (is it through translation and back-translation?) How HP were recruited? Based on what criteria?

Line 205: “…were conducted online” how? Via Zoom? Via another software, etc.?

Line 211: “During the qualitative study, we discovered how the subject of extending the screening offer if currently at the centre of debates in France”. At what point of time during the qualitative study? During the tata analysis? Data collection?

Also please replace discovered with noted.

While authors divided their sample into health professionals and scientists, they later noted that 16 out of 17 are HPs and researchers at the same time and 1 is a medical biologist. So why is this dichotomy necessary and how it plays when interpreting data, if any? Also, what is the difference between scientist and researcher and why not stick to one term scientist or researcher?

p.11 line 233: this theme is not only about additional training of prescribers but also about the limited number when it comes to human resources (genetic counselors, medical geneticist, etc.) available to counsel pregnant people about the test.

p.12 lines 272-273: the lack. Of harmonisation in antenatal care includes a variety of factor. Is it the consent form? The information offered? The technology used? Please explain further.

p.14 line 318: please nuance how this previous approach considered a medical negligence and whose responsibility is it in this analysis of the interview? And in what way it harms pregnant women? It it because the consent is not informed anymore i.e the pregnant woman is somehow forced to consent? Etc….

p.17 lines 377-378: “…has agreed on a list of incidental finding classified as valid” please explain. What does valid imply?

p.25 lines 570-571: How the terms standard vs. expanded are defined? Based on what criteria?

Please explain.

p.25 line 587-589: what is a generic consent form and how is it different from other consent forms?

p.25-26 lines 589-593: “While it is important to provide detailed information, an overload of information should be avoided as this could undermine rather than facilitate the ability of pregnant women to make autonomous choices….decision-making process”: how can this be done? How should a HP provide detailed information while avoiding an overload of information? The sentence structure and meaning are confusing.

What are the limitations and strengths of your study? Did you notice any difference in between for example the perceptions of medical geneticists and those of obgyns?

A reference that could be of interest to interpret and discuss your data:

Haidar, H., Birko, S., Laberge, AM. et al. Views of Canadian healthcare professionals on the future uses of non-invasive prenatal testing: a mixed method study. Eur J Hum Genet (2022). https://doi.org/10.1038/s41431-022-01151-5

Reviewer #3: I read with great interest the Manuscript titled “Health professionals and scientists’ views on genome-wide NIPT in the French public health system: Critical analysis of the ethical issues raised by prenatal genomics” (PONE-D-22-21988), which falls within the aim of this Journal.

In my honest opinion, the topic is interesting enough to attract the readers’ attention. Nevertheless, authors should clarify some point and improve the discussion citing relevant and novel key articles about the topic.

Authors should consider the following recommendations:

- Inclusion/exclusion criteria should be better clarified.

- The authors have not adequately highlighted the strengths and limitations of their study. I suggest clarifying these points.What are the actual clinical implications of this study? it is important to report the results obtained by the authors in the context of clinical practice and to adequately highlight what contribution this study adds to the literature already existing on the topic and to future study perspectives.

- Recent and novel evidence suggested that epigenetic changes, in particular altered expression of selective miRNA, may play a key role in both placental-induced diseases such intrauterine growth restriction. It would be mandatory to discuss (at least briefly) this topic, referring to: PMID: 28466013; PMID: 20104830.

- The real challenge in the era of molecular medicine is to find a biomarker, or even better a panel of biomarkers, for early diagnosis of pre-eclampsia, intrauterine growth restriction and stillbirth. I would stress this point, referring to: PMID: 28243732; PMID: 35245721.

6. PLOS authors have the option to publish the peer review history of their article (what does this mean?). If published, this will include your full peer review and any attached files.

Reviewer #1: No

Reviewer #2: **Yes: **Hazar Haidar

Reviewer #3: **Yes: **Pietro Serra

---

## [Author Response · Author response to Decision Letter 0]

22 Sep 2022

Dear Reviewers,

Thank you very much for reviewing our paper. We very much appreciated receiving your comments, questions and suggestions on how to improve our paper.

We have deposited our research data for this paper with the UK Data Service on 12 September 2022. The data collection are currently under review and will be available at this link very soon: https://doi.org/10.5255/UKDA-SN-855970

Reviewer #1: 

1. ‘I would suggest to try to ameliorate the presentation of the result in a more schematic way, moreover to add a table with a summary of findings’

Thanks for this excellent suggestion, we have added a table just below the abstract.

2. ‘I would recommend to go even more thought the crucial role of women awareness of the meaning of this test, is has been demonstrated worldwide that women understanding of this test is still quite confusing, the words diagnostic and screening are still often confused this pose enormous risk for health care professional, for this point I suggest to read and cite: PMID: 33111167 Quaresima P, Visconti F, Greco E, Venturella R, Di Carlo C. Prenatal tests for chromosomal abnormalities detection (PTCAD): pregnant women's knowledge in an Italian Population. Arch Gynecol Obstet. 2021 May;303(5):1185-1190. doi: 10.1007/s00404-020-05846-2. Epub 2020 Oct 27. PMID: 33111167.’

Thank you for your comment. We mentioned in the discussion section the importance of genetic counselling to support women's understanding of the difference between screening and diagnosis. Thanks for the reference, we have added it.

This part of the article also addresses the issue of women's understanding of the tests and the impact that can be generated by the offer of a new test that women had not heard of before: ‘However, giving women the option to consent or not to the search for other chromosomal abnormalities can come as a surprise or an unexpected choice, as described by this midwife.’

Finally, in this paper, we also questioned the women's perspective even if it is not the main focus.

Reviewer #2: 

1. ‘For the methodology, the authors state that they analyse the views of French HPs around GW-NIPT, based on semi-structured interviews and a comprehensive literature review that includes clinical consent forms, guidelines, regulatory texts, institutional reports, laboratory information leaflets. While they report their findings from the qualitative data, the comprehensive review part is missing. So, it is not clear if the authors conducted two separated studies (empirical and conceptual analysis based on comprehensive review) and that they are trying to combine in one study and in such case, they should describe how the comprehensive review was performed, the inclusion and exclusion criteria, what kind of studies were included, etc. or if they are referring to the literature from diverse sources to better interpret their findings.’

Thanks for your comments on the methodological part. We have added two paragraphs at the beginning of the methodological section, which explain the articulation between the documentary analysis and the empirical analysis, and then how the literature review was carried out. We also specified the inclusion and exclusion criteria for the study.

2. ‘The translation of the quotes needs to be revised as some sections seem to reflect a literal translation and the written English should conform with the academic English writing throughout the manuscript. Many sentences are structurally unclear, rendering therefore the meaning and the understanding very complex and unachievable.’ 

Thank you for your very careful proofreading. We have addressed the translation and written English problems throughout the article.

3. ‘Introduction p.4 line 73: The authors state “despite clinical benefits of GW-NIPT” without mentioning what are these. Please nuance.’

Thanks, we have modified this passage in writing 'some potential benefits' and specified in the introduction what they are.

4. ‘Line 74: this screening test is largely driven by industry. So what does this mean and how it does impact its use and implementation?’

We have deleted this section because we lack of evidence to suggest how this technology is driven: by the industry or the science evidence. This understanding would require further research and investigation, and would involve focusing on this topic, which is not the objective of this article.

5. ‘Context Why not give context about TOP in France? Is it legal? Prohibited at some point in time during the pregnancy? The authors only briefly discuss TOP as a theme that was developed in their interviews. It would be of benefit to the reader to have a glance about the situation of pregnancy termination in France.’

Thanks for this excellent suggestion, we have added the French legal context to better understand the context of expanded NIPT offer.

6. ‘Methodology Lines 201-204: Many details are missing from the methodology used to collect the data. For instance, how the themes were developed and how were they validated? In what language were the interviews conducted? How the translation into English was done and validated (is it through translation and back-translation?) How HP were recruited? Based on what criteria?

Line 205: “…were conducted online” how? Via Zoom? Via another software, etc.?

Line 211: “During the qualitative study, we discovered how the subject of extending the screening offer if currently at the centre of debates in France”. At what point of time during the qualitative study? During the tata analysis? Data collection?

Also please replace discovered with noted.

While authors divided their sample into health professionals and scientists, they later noted that 16 out of 17 are HPs and researchers at the same time and 1 is a medical biologist. So why is this dichotomy necessary and how it plays when interpreting data, if any? Also, what is the difference between scientist and researcher and why not stick to one term scientist or researcher?’

Thank you for this very useful feedback. We have addressed the main methodological shortcomings and added elements that needed to be clarified.

We have simplified and retained the category of "scientist" to better understand who we are talking about. This category includes people who conduct research on NIPT. Some health professionals are also scientists (11 of the 17 interviewed) and have published work on NIPT, which we feel is important to note. 

7. ‘p.11 line 233: this theme is not only about additional training of prescribers but also about the limited number when it comes to human resources (genetic counselors, medical geneticist, etc.) available to counsel pregnant people about the test.’

Yes, we agree, we have amended the title. We have also chosen to use the terminology of ‘Health professionals’ to refer to ‘prescribers’, as the role of prescribers is broader.

8. ‘p.12 lines 272-273: the lack. Of harmonisation in antenatal care includes a variety of factor. Is it the consent form? The information offered? The technology used? Please explain further.’

In the following paragraph, we outline the different factors related to information and the consent forms that are proposed.

9. ‘p.14 line 318: please nuance how this previous approach considered a medical negligence and whose responsibility is it in this analysis of the interview? And in what way it harms pregnant women? It it because the consent is not informed anymore i.e the pregnant woman is somehow forced to consent? Etc….’

Thank you for your very interesting comment, we have indeed revised this part.

10. ‘p.17 lines 377-378: “…has agreed on a list of incidental finding classified as valid” please explain. What does valid imply?’

This is the category used in the Belgian Guidelines. However, we have clarified what they mean by valid: ‘technically and with validated evidence on the associated phenotype’.

11. ‘p.25 lines 570-571: How the terms standard vs. expanded are defined? Based on what criteria? Please explain.’ 

We use the terminology of ‘expanded NIPT’ to dialogue with other scientific works because these are the categories used. See for example: Kater-Kuipers A, de Beaufort ID, Galjaard RJ, Bunnik EM. Ethics of routine: a critical analysis of the concept of ‘routinisation’in prenatal screening. Journal of Medical Ethics. 2018 Sep 1;44(9):626-31. Christiaens L, Chitty LS, Langlois S. Current controversies in prenatal diagnosis: Expanded NIPT that includes conditions other than trisomies 13, 18, and 21 should be offered. Prenatal Diagnosis. 2021;41(10):1316-23.

‘Standard NIPT’ is a terminology that we propose here in order to be able to differentiate the scope of the NIPT offer between the offer for the three trisomies (‘standard’) and the extended offer (for other abnormalities).

12. ‘p.25 line 587-589: what is a generic consent form and how is it different from other consent forms?’

Thanks. We have removed the word 'generic' as this was our analysis and not the word used by the laboratory to describe how consent is obtained.

13. ‘p.25-26 lines 589-593: “While it is important to provide detailed information, an overload of information should be avoided as this could undermine rather than facilitate the ability of pregnant women to make autonomous choices….decision-making process”: how can this be done? How should a HP provide detailed information while avoiding an overload of information? The sentence structure and meaning are confusing.’

Thank you to pointing this out. We have revised this sentence.

14. ‘What are the limitations and strengths of your study?’ 

Thanks for this comment. We have added a table on the contributions of the study between the abstract and the introduction, and a paragraph on the limitations after the conclusion.

15. Did you notice any difference in between for example the perceptions of medical geneticists and those of obgyns?

Thanks. We did not notice any particular difference. In our opinion the difference is rather between the divisions around the test: some health professionals being rather in favour or rather against its offer.

16. A reference that could be of interest to interpret and discuss your data:

Haidar, H., Birko, S., Laberge, AM. et al. Views of Canadian healthcare professionals on the future uses of non-invasive prenatal testing: a mixed method study. Eur J Hum Genet (2022). https://doi.org/10.1038/s41431-022-01151-5

Thanks a lot for this very interesting reference. We quoted it in our article.

Reviewer #3: 

1. Inclusion/exclusion criteria should be better clarified.

Thank you for this comment, we have clarified this point in the methodology section.

2. The authors have not adequately highlighted the strengths and limitations of their study. I suggest clarifying these points.

Thanks for this suggestion. We have added a table on the contributions of the study between abstract and the introduction, and a paragraph on the limitations after the conclusion.

3. What are the actual clinical implications of this study? it is important to report the results obtained by the authors in the context of clinical practice and to adequately highlight what contribution this study adds to the literature already existing on the topic and to future study perspectives.

Thanks for this comment.

For the concrete clinical implications, this implies addressing three ethical issues that we highlight throughout the article: the differing quality of counselling, the complexity of achieving informed consent, and the avoidance of harm to pregnant women.

4. Recent and novel evidence suggested that epigenetic changes, in particular altered expression of selective miRNA, may play a key role in both placental-induced diseases such intrauterine growth restriction. It would be mandatory to discuss (at least briefly) this topic, referring to: PMID: 28466013; PMID: 20104830. https://pubmed.ncbi.nlm.nih.gov/28466013/

The real challenge in the era of molecular medicine is to find a biomarker, or even better a panel of biomarkers, for early diagnosis of pre-eclampsia, intrauterine growth restriction and stillbirth. I would stress this point, referring to: PMID: 28243732; PMID: 35245721. https://pubmed.ncbi.nlm.nih.gov/28243732/

Thank you for these very specific medical suggestions, but we do not offer medical analysis in this paper as we do not have the expertise. Our approach is at the intersection between sociology, bioethics and law/social policy.

---

## [Decision Letter · Decision Letter 1]

19 Oct 2022

Health professionals and scientists’ views on genome-wide NIPT in the French public health system:

Critical analysis of the ethical issues raised by prenatal genomics.

PONE-D-22-21988R1

Dear Dr. Perrot,

We’re pleased to inform you that your manuscript has been judged scientifically suitable for publication and will be formally accepted for publication once it meets all outstanding technical requirements.

Kind regards,

Antonio Simone Laganà, M.D., Ph.D.

Academic Editor

PLOS ONE

Additional Editor Comments (optional):

Authors performed the required corrections, which were positively evaluated by the reviewers. I am pleased to accept this paper for publication.

Reviewers' comments:

Reviewer's Responses to Questions

**Comments to the Author**

1. If the authors have adequately addressed your comments raised in a previous round of review and you feel that this manuscript is now acceptable for publication, you may indicate that here to bypass the “Comments to the Author” section, enter your conflict of interest statement in the “Confidential to Editor” section, and submit your "Accept" recommendation.

Reviewer #3: All comments have been addressed

2. Is the manuscript technically sound, and do the data support the conclusions?

Reviewer #3: Yes

3. Has the statistical analysis been performed appropriately and rigorously? 

Reviewer #3: N/A

4. Have the authors made all data underlying the findings in their manuscript fully available?

Reviewer #3: Yes

5. Is the manuscript presented in an intelligible fashion and written in standard English?

Reviewer #3: Yes

6. Review Comments to the Author

Reviewer #3: I read with great interest the Manuscript titled Health professionals and scientists’ views on genome-wide NIPT in the French public health system: Critical analysis of the ethical issues raised by prenatal genomics." (PONE-D-22-21988R1), which falls within the aim of this Journal. In my honest opinion, the topic is interesting enough to attract the readers’ attention.

7. PLOS authors have the option to publish the peer review history of their article (what does this mean?). If published, this will include your full peer review and any attached files.

Reviewer #3: **Yes: **Pietro Serra

---

## [Editor Report · Acceptance letter]

24 Oct 2022

PONE-D-22-21988R1 

Health professionals and scientists’ views on genome-wide NIPT in the French public health system: Critical analysis of the ethical issues raised by prenatal genomics 

Dear Dr. Perrot:

I'm pleased to inform you that your manuscript has been deemed suitable for publication in PLOS ONE. Congratulations! Your manuscript is now with our production department. 

Kind regards, 

on behalf of

Dr. Antonio Simone Laganà 

Academic Editor

PLOS ONE